# A Comprehensive Review of the Bovine Immune Response to Pathogens

**DOI:** 10.3390/ijms26178461

**Published:** 2025-08-30

**Authors:** Ana Lesta, Pablo Jesús Marín-García, Lola Llobat

**Affiliations:** 1Molecular Mechanisms of Zoonotic Diseases (MMOPS) Research Group, Departamento Producción y Sanidad Animal, Salud Pública y Ciencia y Tecnología de los Alimentos (PASAPTA), Facultad de Veterinaria, Universidad Cardenal Herrera-CEU, CEU Universities, 46113 Valencia, Spain; ana.lesta@alumnos.uchceu.es; 2Departamento Producción y Sanidad Animal, Salud Pública y Ciencia y Tecnología de los Alimentos (PASAPTA), Facultad de Veterinaria, Universidad Cardenal Herrera-CEU, CEU Universities, 46113 Valencia, Spain; pablo.maringarcia@uchceu.es

**Keywords:** *Bos taurus*, bovine, immune system, pathogens, response

## Abstract

Dairy cattle are constantly exposed to a wide range of pathogens, which can produce substantial economic losses. The maintenance of homeostasis is not only dependent on the intrinsic characteristics of the animals but also on environmental factors such as the productive system, heat stress, and exposure to vectors and contaminated pastures. In this context, the bovine immune system plays a critical role in maintaining health and productivity. This review provides a comprehensive overview of both innate and adaptive responses in cattle, remarking on key components and summarizing the normal immune response against some of the most frequent pathogens in bovines, as well as how these pathogens have developed strategies to evade or modulate the host’s immune system. A deeper understanding of these mechanisms is essential for improving therapeutic strategies and disease prevention in livestock production.

## 1. Introduction

Milk production plays a crucial role in global food security and economic sustainability, providing essential nutrition and livelihoods for millions, particularly in developing countries [1]. The dairy industry also contributes significantly to agricultural economies, with increasing demand driven by population growth, urbanization, and dietary shifts [2]. In 2024, global milk production experienced modest growth, continuing trends observed in previous years. In 2023, unprocessed milk production increased by 2.1%, reaching 964 million tons, with cow’s milk accounting for 81.1%. This growth was slightly below the compounded average growth rate of 2.2% observed from 2015 to 2023 [3]. The OECD-FAO Agricultural Outlook projects that world milk production will grow at an annual rate of 1.7%, reaching 1020 million tons by 2030 [3].

However, this sector is facing biological challenges; dairy cows are constantly exposed to a wide range of pathogens, which may significantly impact animal health, productivity, and food safety. The immune system plays a pivotal role in maintaining health and defending organisms against a wide range of pathogens. Although the fundamental mechanisms of immune responses are conserved across mammalian species, most of our current understanding derives from studies in humans and laboratory mice. This bias has created a knowledge gap regarding immune function in other species, including cattle, which are not only economically important but also exhibit unique immunological features.

Bovines possess several distinctive traits within both their innate and adaptive immune systems that reflect evolutionary adaptations to their environments and pathogen exposures. Notably, they have a higher proportion of γδ T cells in peripheral blood, a unique composition of Toll-like receptors (TLRs), and specific adaptations in the expression of acute-phase proteins and antigen-presenting molecules [4,5,6]. These differences underscore the necessity of species-specific research to better understand bovine immunity and improve disease management, vaccination strategies, and overall animal welfare.

This review focuses on the general principles of immune response, highlighting key components and pathways of both the innate and adaptive immune systems. Particular attention is given to bovine-specific mechanisms, variations, and immune responses to pathogens. By emphasizing findings derived from bovine research, this work aims to contribute to a more complete and accurate representation of mammalian immunology beyond traditional model organisms.

## 2. Bovine Immune System

Although most immune mechanisms are similar between mammals, most of the current information related to immune response comes from studies in humans and mice, so this section will focus on general principles while highlighting specific findings in bovines.

### 2.1. Innate Immune Response

Innate immunity serves as the first line of defense, providing rapid recognition and response to pathogens. The innate immune response involves three relevant stages. Central to this system are Pattern Recognition Receptors (PRRs), including Toll-like receptors (TLRs), NOD-like receptors (NLRs), and RIG-I-like receptors (RLRs). First, TLRs (TLR1–TLR9) are expressed in macrophages, dendritic cells, and epithelial cells, detecting bacterial, viral, and parasitic components, while NLRs (e.g., NOD1, NOD2, NLRC4, NLRP3) mediate intracellular pathogen sensing and inflammasome formation. RLRs, including RIG-I and MDA5, recognize viral RNA within the cytoplasm, initiating antiviral responses. Second, inflammasomes such as NLRP3 and AIM2 play key roles in inflammation by activating caspase-1, leading to the secretion of IL-1β and IL-18, and detecting cytosolic DNA, respectively. Complementing these pathways, acute phase proteins (APPs) including haptoglobin, serum amyloid A, fibrinogen, C-reactive protein, and alpha-1-acid glycoprotein are predominantly produced by the liver during infection, contributing to pathogen opsonization, complement activation, and modulation of inflammatory responses. Finally, cytokines orchestrate innate immune signaling, with proinflammatory mediators (TNF-α, IL-1β, IL-6, IL-12, IFN-γ) promoting pathogen clearance, and anti-inflammatory cytokines (IL-10, TGF-β) maintaining immune homeostasis. Phagocytic cells—macrophages, neutrophils, and dendritic cells—play complementary roles, from direct pathogen engulfment and reactive species production to antigen presentation and activation of adaptive immunity. Innate lymphoid cells (ILCs) and natural killer (NK) cells further enhance host defense. Classical NK cells mediate cytotoxicity and secrete IFN-γ, while bovine NKT cells, including WC1+ γδ T cells, bridge innate and adaptive immunity. Together, these cellular and molecular components establish a dynamic and efficient innate immune network, enabling bovines to detect, respond to, and control diverse pathogens.

The innate immune response involves three relevant stages. First, the recognition of pathogen-associated molecular patterns (PAMPs) on the surface of the pathogens that enter the organism. Second, the activation of immune cells and the following release of all kinds of inflammatory mediators. Finally, the effector phase, where phagocytes and natural killer (NK) cells eliminate the pathogens. Simultaneously, the inflammatory response is initiated, which recruits additional immune cells to the site of infection [7]. Therefore, the immune response involves a cascade of mechanisms that implies a complex interaction between cellular and humoral components, both supporting the protection of the animal. For example, epithelial cells, which form the first line of defense, contribute to innate immunity as physical barriers in the respiratory, gastrointestinal, and urogenital tracts [8]. In addition, these cells further enhance this defense by secreting antimicrobial peptides and defensins, among other components [9]. Other cell components included in the innate immune system are neutrophils, natural killer cells (NK), dendritic cells (DCs), gamma delta T cells (γδT), mucosa-associated invariant T cells (MAITs), macrophages (Mϕ) and granulocytes. These cells work together to achieve a quick and non-specific defense against pathogens. In addition, the bovine innate immune system utilizes pattern recognition receptors (PRRs) as Toll-like receptors (TLRs) to detect PAMPs. The activation of these receptors is followed by the production of cytokines and chemokines, which help to coordinate the immune response [10]. Table 1 shows some cytokines and their functions.

The innate immune response in cattle can vary between individuals and can be influenced by several factors such as genetics, age, nutrition or environmental stressors. This variation affects disease susceptibility, as seen in mastitis and other pathologies [11], and it has been observed in cytokine responses to various pathogen-associated molecular patterns (PAMPs), including bacterial lipopolysaccharides (LPS) and viral mimics such as R848 [10]. In response, the bovine immune system has evolved specific adaptations to fight common pathogens.

**Table 1 ijms-26-08461-t001:** Main cytokines and their functions in bovines [12]. IL: interleukin; APP: acute phase protein; IFN: interferon; NK: natural killer; MHC: Major Histocompatibility Complex; Ig: immunoglobulin; TNF: Tumor necrosis factor; DC: Dendritic cell; MCP: Monocyte Chemoattractant Protein; TGF: Transforming Growth Factor.

Cytokines	Function
IL-1α, IL1-β	Proinflammatory Leukocyte adhesion and migration, APPs induction Promotes Th2 cytokine and IFN-γ production
IL-2	B and T cell proliferation Enhance NK cell function
IL-4	B and CD8+ cell proliferation, Th2 response activation Enhances MHC-II expression Stimulates IgE and IgG production
IL-5	B cell proliferation and maturation Stimulates IgA and IgM production
IL-6	Proinflammatory Th2 response, Th17 cells, APPs induction, B cell differentiation into plasmatic cells, IgA production Anti-inflammatory: inhibits some activities of TNF-α and IL-1, promotes IL-10
IL-7	B and T cell growth factor
IL-8	Proinflammatory
IL-10	Anti-inflammatory (inhibits proinflammatory cytokines) Suppresses macrophages and DCs
IL-11	Proinflammatory APPs induction
IL-12	Anti-inflammatory NK cell and phagocyte activation Th1 response activation
IL-17A-F	Proinflammatory Cytokine/chemokine inhibitors (IL-1, -6, -8, -21, TNF-β, MCP-1)
IFN-α/β	Proinflammatory Antiviral activity, NK cell activation
IFN-γ	Proinflammatory Antiviral activity, macrophage and Th1 response activation Increases neutrophil/monocyte function and MHC-I and MHC-II expression Inhibits Th2 response, suppresses IgE production
TGF-β	Immunosuppressor, Treg induction Inhibits lymphocyte and macrophage production. Promotes Th17 differentiation
TNF-α/β	Proinflammatory Neutrophil, macrophage and lymphocyte activation, APPs induction IL-1, IL-6, IL-8 production
Chemokines	Regulate leukocyte recruitment and migration

#### 2.1.1. Pathogen Recognition

Pattern Recognition Receptors (PRRs), including Toll-like receptors (TLRs), NOD-like receptors (NLRs), and RIG-I-like receptors (RLRs) are central components of the innate immune system. TLRs play an important role in pathogen recognition. These proteins, located on the surface of different immune cells, can detect PAMPs. When TLRs are activated, they start a chain reaction inside the cell that ends with the production of inflammatory molecules, such as cytokines [13,14]. There have been ten functional TLRs identified in bovines (TLR1 to TLR10), which are expressed in macrophages, dendritic cells, and epithelial cells, detecting bacterial, viral, and parasitic components, and which have evolved to recognize specific PAMPs. For example, TLR2 forms heterodimers with TLR1 to detect lipopeptides from Gram-negative bacteria, and with TLR6 to detect lipopeptides, peptidoglycans, and other components from Gram-positive bacteria [15,16]. TLR3 plays a significant role in recognizing viral double-stranded RNA, as in bovine intestinal epithelial cells [17]. Polymorphisms in the TLR4 gene, which play a crucial role in recognizing Gram-negative bacterial lipopolysaccharides, have been linked to differences in inflammatory responses and resistance to intramammary infections [18]. TLR7/TLR8 (which recognize R848) are involved in the detection of viral single-stranded RNA, mainly in plasmacytoid dendritic cells [19]. Polymorphisms in TLR7/8 genes can result in differential activation of downstream signaling pathways, leading to variation in cytokine production, such as TNF-α, IL-6, and IFN-α, which influence susceptibility to bacterial and viral infections [20,21]. TLR5 detects bacterial flagellin and TLR9 detects unmethylated CpG, which is frequently present in bacterial and viral DNA [22]. Finally, TLR10, the most recently described, may have both pro- and anti-inflammatory effects [23].

There are other innate receptors involved in this process: NOD-like receptors (NLRs) and retinoic acid-inducible gene I (RIG-I)-like receptors (RLRs). NLRs are proteins present in macrophages, dendritic cells, and epithelial cells. These are found exclusively in the cytosol and the nucleus, facilitating the detection of foreign nucleic acids and other intracellular components inside the cells. NOD1 and NOD2 can activate the NF-κB pathway through RIP2/RICK, which stimulates the production of proinflammatory cytokines such as IL-1β and IL-18 [24]. However, some NLRs as NLRC3 and NLRP2/4 inhibit this pathway, modulating the expression of TNF-α y TRAF6 [25,26]. RLRs can detect foreign RNA within the cytoplasm, initiating signaling and activating transcription factors such as NF-κB and IRF-3 (interferon regulatory factor 3) [27]. In bovines, NOD2, a key NLR, and two RLRs, RIG-I and MDA-5, have been identified as critical cytosolic sensors of viral RNA. NOD2 functions as a cytosolic receptor for single-stranded viral RNA [28], while RIG-I and MDA-5 are RNA helicases that recognize viral RNA in the cytoplasm. Upon activation, RIG-I and MDA-5 signal through the mitochondrial antiviral-signaling protein (MAVS) to induce downstream activation of IRF3/7 and the subsequent production of type I interferons, thereby initiating a potent antiviral immune response [29].

Figure 1 represents a diagram of a bovine immune cell showing key TLRs (TLR2/1, TLR2/6, TLR3, TLR4, TLR5, TLR7/8, TLR9, TLR10) on the cell surface or endosomes.

#### 2.1.2. Local Inflammation and Cellular Recruiting

Inflammation is a key process within the innate response that can be triggered by the complement system or by immune cells, such as macrophages, phagocytes or innate lymphocytes, in the event of tissue damage. During acute inflammation, affected tissues become swollen and painful, due to the arrival of immune cells and plasma proteins to the site of injury [8].

Inflammasomes are multiprotein complexes triggered by a variety of stimuli associated to tissular damage or stress, such as the presence of PAMPs in the cytoplasm, toxins, or reactive oxygen species (ROS). This activation results in the liberation of proinflammatory cytokines such as IL-1β and IL-18, activating local inflammation [30]. The activation of the inflammasomes can differ across species. For example, in rodents, NLRP1 inflammasome is activated by pathogens like *Toxoplasma gondii* or by *Bacillus anthracis* lethal toxin (LT), leading to pyroptosis in macrophages, which is a highly inflammatory form of programmed cell death. In contrast, bovine NLRP1 inflammasome does not respond to this toxin, and its macrophages undergo a slower and non-pyroptotic death [31]. In another study, it was observed that pyroptosis can be triggered in bovine macrophages while infected by *Mycobacterium bovis*. NLRP3 inflammasome is activated, inducing IL-1β maturation, ASC oligomerization, and caspase-1 activation [32].

Acute-phase proteins (APPs) are significant participants in the innate immune response. Their production in the liver is stimulated through IL-1β, IL-6, and TNF-α, in response to infection, inflammation, or injury [33]. These proteins include C-reactive protein (CRP), serum amyloid A (SAA), and haptoglobin (Hp); these increase dramatically during acute inflammation, so these proteins are common indicators of inflammation, infection, or tissue damage [34,35,36]. In cattle, the major acute phase proteins are SAA and Hp; these are widely used in diagnosis and are normally found in serum or plasma, but they can also be found in milk, saliva, and meat juice [33]. Although acute phase proteins (APPs) are typically synthesized in the liver, research has indicated that they are also expressed locally in the bovine forestomach and abomasum. This expression suggests a potential role in local immune regulation within the gastrointestinal mucosa [37]. It has been shown that APPs responses are variable and depend on the nature of the agent, among other factors. For instance, viral infections tend to induce a milder APP response in comparison to bacterial infections [34].

The complement system is a mechanism that consists of over 30 plasma proteins that provide a cascade-like defense against pathogens in both innate and adaptive responses [35]. These proteins are produced primarily by the liver, but also by a variety of cell types and tissues [38,39]. The complement system can be activated through three pathways: classical, alternative, and lectin pathways [40,41]. The classical pathway is normally initiated by antigen-antibody complexes, but also viral envelopes, Gram-negative bacterial walls, C-reactive protein, cytoskeletal intermediate filaments, and central nervous system myelin [41]. The alternative pathway can be activated by spontaneous hydrolysis or by the presence of foreign components. This pathway is particularly important in cattle because the kinetics of activation differs from that of mice and humans; this difference may be associated to conglutinin, which acts as a stabilizing control protein for the ruminant C3-convertase C3bBb [42]. Finally, the lectin pathway has a complex activation mediated by a variety of proteins, such as mannose-binding lectin (MBL), that recognize foreign carbohydrates.

These three pathways converge on a terminal pathway, leading to the formation of the membrane attack complex (MAC) composed from C5 to C9. This complex can cause cell lysis by perforating cell membranes [41]. In addition to its lytic function, the complement system generates components that promote inflammation, opsonization and chemotaxis, but also contributes to the clearance of immune complexes and apoptotic cells. For example, C3b and C3bi enhance phagocytosis in bacteria, while C5 acts as a potent chemoattractant [43].

#### 2.1.3. Effector Phase

Phagocytosis is one of the most efficient processes displayed during the innate response. This process is mediated by neutrophils and macrophages: neutrophils are present in the peripheral blood and are often the first cell type that arrives at the site of infection; macrophages are mostly found on other tissues and can be modulated by host-derived factors, such as epithelial cell secretomes, to enhance their microbicidal functions [44].

These cells can detect and destroy foreign particles, including bacteria or damaged cells. Once they catch them, they form a vesicle called phagosome, which later fuses with lysosomes to degrade what has been internalized [8,45]. In cattle, phagocytosis becomes more efficient when opsonization takes place. This process involves immunoglobulins (IgG), which cover the surface of the pathogen and help immune cells recognize it more easily, “marking” the pathogen and, therefore, facilitating phagocytosis [10]. Phagocytes can also secrete proinflammatory molecules that activate and attract more cells to the site affected. They can release reactive oxygen species (ROS) and antimicrobial peptides, which directly attack pathogens [10]. Another mechanism adapted to trap and kill pathogens is that shown by neutrophils: a process known as NETosis. This involves the release of DNA, histones, and antimicrobial proteins to form extracellular trap structures (NETs) to restrain pathogens [46]. *Mycoplasma bovis*, an important bovine respiratory and mastitis pathogen, has been shown to induce NETs only when inactivated, suggesting an active evasion mechanism by live bacteria [47].

Bovine Dendritic Cells (DCs) are other classes of phagocytic cells, quite efficient in the detection of pathogens and the activation of T lymphocytes. These are present in different tissues and express many receptors that allow them to detect viruses, bacteria, and parasites [48]. When dendritic cells are present in tissues, they remain in an “immature” stage, in which they have a high capacity to internalize the pathogens they recognize. When they phagocytose these pathogens, the cells enter a “mature” stage, in which their phagocytic capacity is severely reduced [49]. At this point, they begin to degrade the pathogens and leave the tissues following the lymphatic circulation until they reach the secondary lymphoid organs. Here they present small pathogenic peptides to T lymphocytes for its activation [50]. Therefore, DCs represent the transition between innate and adaptive responses. It has been studied that lactoferrin, a protein produced by mucosal epithelial cells and present in different secretions like milk, enhances TLR-7 response in plasmacytoid DCs, producing significant amounts of interferon alfa (IFN-α) [51]. Plasmacytoid DCs are cells specialized in the production of type I IFN. These proteins can interfere with viral replication, and are subdivided into IFN-α, produced by plasmacytoid DCs and macrophages, and IFN-β, produced by many cell types. All interferons bind to a specific receptor expressed on the surface of a wide variety of cell types throughout the body. When virus-infected cells produce type I IFN, these trigger a cellular response that slows down viral replication inside the cells [52].

Innate lymphoid cells (ILC) are lymphocytes lacking specific antigenic receptors, so the activation depends entirely on innate receptors or cytokines. These are in lymphoid tissues and include: ILC1, ILC2, and ILC3, present in different tissues; natural killer (NK) cells, found especially in blood, and lymphoid tissue inductors (LTi). Bovine NK cells, or NKp46/NCR1+ CD3− cells, can be subdivided into two subsets: CD2+ and CD2−/low cells, present in peripheral blood and lymph nodes, respectively [53].

Both ILC1 and NK cells are linked to type I response (macrophage activation, cytotoxicity) and produce IFN-γ, a type II IFN that regulates the response to intracellular pathogens. ILC2 and ILC3 are active in type II (alternative macrophage activation) and III (phagocytosis, antimicrobial peptides) responses, respectively. ILC2 produces IL-4, IL-5, and IL-13 in response to extracellular parasites and is involved in allergic responses, while ILC3 produces IL-17 and IL-22, regulating the activation of Th17 lymphocytes [54].

Interestingly, bovines have an increased proportion of γδ T cells in comparison to mice and humans, which improves their immune surveillance and early pathogen recognition. These are a population of T lymphocytes present in peripheral blood that produces a rapid, non-Major Histocompatibility Complex (MHC) dependent response, followed by the production of IFN-γ and IL-17 [4]. This mechanism suggests a role in mucosal defense and early pathogen control. Interestingly, the highest frequency of γδ T cells occurs during the neonatal period, and it declines as the animal ages, which suggests that it may be particularly important in the immune response of younger cattle [55].

Figure 2 summarizes different pathogen recognition by innate receptors.

### 2.2. Adaptive Immune Response

Unlike the innate immune response, which is immediate and non-specific, the adaptive response produces a highly specialized defense, characterized by recognizing specific pathogens and generating a targeted response, as well as developing immune memory for faster recognition upon following exposures. Adaptive immunity in bovines provides highly specific and long-lasting protection against pathogens through coordinated cellular and humoral responses. Key lymphocytes include CD4+ helper T cells, CD8+ cytotoxic T cells, and γδ T cells, particularly WC1+ subsets unique to cattle, which bridge innate-like and adaptive functions. B cells produce a variety of immunoglobulins (IgM, IgG1, IgG2, IgA, IgE, IgD) that mediate pathogen neutralization, complement activation, and long-term immunity. Antigen presentation is central to adaptive responses. MHC class I molecules present endogenous antigens to CD8+ T cells, while MHC class II molecules present extracellular antigens to CD4+ T cells. Co-stimulatory molecules, including CD80, CD86, and CD40, ensure effective lymphocyte activation, enabling precise and robust immune responses. Cytokines further direct immunity, with Th1-type responses (IFN-γ, IL-2) promoting cell-mediated defense, Th2-type responses (IL-4, IL-5, IL-13) supporting humoral immunity, and regulatory cytokines (IL-10, TGF-β) maintaining immune balance. Bovine immunoglobulins exhibit specialized functions and tissue distributions. IgM mediates primary responses and complements fixation, IgG subclasses support secondary responses and neonatal immunity, IgA protects mucosal surfaces, IgE participates in anti-parasitic defense, and IgD regulates B cell activation. Together, these cellular and molecular components form a sophisticated adaptive immune network that enables cattle to mount highly specific, regulated, and effective responses against diverse pathogens.

This system relies primarily on the activation of T lymphocytes, which produce a cellular response, and B lymphocytes, responsible for humoral immunity. Upon encountering the antigen, T and B cells differentiate and proliferate, creating a subset of effector and memory cells [56]. Antibodies produced by B cells can target specific pathogens for their elimination through opsonization, agglutination or neutralization. The coordination between cellular and humoral responses, along with the cytokine signaling and other mechanisms, ensures the effectiveness of the adaptive response against pathogens, which can adapt through the infection and produce long-term protection.

#### 2.2.1. Antigen Recognition

Antigen-presenting cells (APCs) are specialized immune cells that capture pathogens and foreign components and present antigenic fragments on their surface. These antigens are displayed bound to Major Histocompatibility Complex (MHC) molecules, enabling recognition by T lymphocytes. In cattle, the MHC system is known as the Bovine Leukocyte Antigen (BoLA) [57]. T lymphocytes recognize this antigen-MHC complex through TCRs (T cell receptors), which leads to T cell activation and the initiation of the adaptive response. While DCs, macrophages, B lymphocytes, and activated T cells are the main APCs, there are other cell types, such as thymic epithelial cells and endothelial cells [58]. Antigens must encounter the APCs, B, and T cells in the lymphoid organs depending on the route of entry. For example, antigens present in the bloodstream would find these APCs in the spleen, while those that enter the organism through the epithelium would be carried to the lymph nodes. Antigen recognition is a decisive step in the adaptive system, since this determines whether the adaptive immune response takes place or not.

There are MHC class I (MHC-I) and MHC class II (MHC-II) molecules. MHC-I present endogenous antigens to CD8+ cytotoxic T cells, while MHC-II present extracellular peptides (including pathogens) to CD4+ helper T cells. This division ensures that the immune response is appropriate to each pathogen. Both MHC classes possess great variability and are highly polygenic, which means that there are several versions codified by different genes, and polymorphic [58]. In fact, the genetic variability of bovine MHC (BoLA) genes contributes to individual and breed differences in disease susceptibility and immune response across bovine breeds. For example, the BoLA-DRB3 gene, part of MHC-II, is highly diverse in Holstein cattle [59]. The BoLA-DRB3 gene also shows high allele diversity in Simmental and Simbrah cattle, but also high individual homozygosis. This may limit antigen presentation despite genetic diversity, although signs of positive selection suggest the maintenance of functional diversity [60]. A recent study across 15 cattle populations identified selective sweeps in key regions of the MHC, including BoLA, BoLA-NC1, MIC1, CD244, and GJA5, which suggests that selection has shaped the diversity and function of MHC genes in bovine species, potentially influencing breed-specific immune responses [61]. Another study revealed novel DRB3 alleles and several MHC-I alleles and haplotypes in three breeds of Bos indicus [62].

Other proteins work as antigen-presenting molecules. The CD1 family of proteins has a similar structure to MHC-I and presents antigen lipids to T cells, which makes them relevant in mycobacterial recognition. It has been shown that cattle express CD1a, CD1b, and CD1e, but lack CD1c and functional CD1d genes, which suggests that they do not have classical NKT cells. However, they can present lipid antigens to CD1-restricted T cells [63]. MR1 (MHC related 1) molecules are also like MHC-I and bind to bacterial metabolic derivatives. These metabolic products are recognized by a family of non-conventional T lymphocytes, MAITs (mucosa-associated invariant T-cells). MAIT cells have been identified in cattle, sharing some features with humans and mice. They are within the mucosal tissue and have been implicated in immune response to mastitis and tuberculosis, increasing frequency in milk and the expression of perforin in peripheral blood, respectively [64].

There are certain proteins, called superantigens (SAgs), that do not require presentation to activate T lymphocytes. These microbial proteins interact with the Vβ region of TCR and bond to MHC-II in an unconventional way [65]. This kind of interaction leads to massive T cell activation and cytokine release, contributing to systemic inflammation. In bovines, *Staphylococcus aureus*, which is a major cause of mastitis, has been shown to encode multiple SAgs, allowing the activation of large subsets of T cells via Vβ specific interactions. In fact, some isolates encode enough SAgs to stimulate nearly the entire bovine T cell repertory [66].

#### 2.2.2. Activation and Proliferation of Lymphocytes

Lymphocytes are the main components in adaptive immune response. They can be broadly categorized into T cells, produced by the thymus and responsible for cell-mediated immunity, and B cells, produced in bone marrow, bursa, and Peyer’s patches, which are responsible for humoral immunity. When naïve T and B cells recognize antigens bound to MHC molecules, these immune cells become active, allowing them to proliferate and differentiate into effector cells [7].

T lymphocytes can be divided into subsets according to their T cell receptors (TCRs): αβ T cells, the most abundant lineage in many mammals, and γδ T cells. In cattle, γδ T cells represent a major regulatory subset, which produce IL-10 and suppress CD4+ and CD8+ T cell proliferation [67]. αβ T can be divided into CD4+ helper T cells and CD8+ cytotoxic T cells, which recognize MHC-II and MHC-I molecules, respectively. CD4+ T cells coordinate immune response by producing cytokines and activating other immune cells. These can differentiate into functional subsets (Th1, Th2, Th17, Tfh, Treg) that activate specific immune responses depending on the cytokines produced, as shown in Table 2. CD8+ differentiate into cytotoxic T lymphocytes (CTLs), which directly kill infected cells. They are essential for eliminating virus-infected cells [7]. A study shows distinct cytokine mRNA profiles in bovine CD4+ and CD8+ T cells: CD4+ cells express a broader range, including IL-4, IL-5, IL-10, and IL-13, while CD8+ cells mainly express proinflammatory cytokines and IFN-γ upon stimulation [68].

**Table 2 ijms-26-08461-t002:** Main Th (CD4+) cell subtypes and their cytokines in bovines. IL: interleukin; IFN: interferon; TNF: Tumor Necrosis Factor; TGF: Transforming Growth Factor.

Subtype	Differentiation Induction	Cytokines Produced	Function/Activation	References
Th1	IL-12, IFN-γ	IFN-γ, IL-2, TNF-α	Macrophages, CD8+	[69,70]
Th2	IL-4	IL-4, IL-5, IL-13, IL-6, IL-10, TNF-α, IL-9, IL-2	Eosinophils, mastocytes, B cells	[69,70]
Th17	TGF-β, IL-6	IL-17A, IL-17F, IL-22, IL-21	Neutrophils	[69,70]
Tfh	IL-21	IL-21	B cells, Antibodies	[69]
Treg	TGF-β	IL-10, TGF-β, IL-35	Immunomodulation CD25 and FoxP3 expression	[69]

There are other lymphocyte phenotypes following the CD (Cluster Differentiation) designation. This nomenclature relies on the protein molecules expressed on the surface of both T and B cells. For example, CD3 is a protein complex found in all T cells, acting as a universal T cell marker, while CD45 comprises a large family of proteins required for T cell signaling. These surface proteins act as regulators of lymphocyte function, which consist mostly of cytokine (CD), antibody (FcR), and complement (CR1 to CR4) receptors [7]. The majority of bovine γδ T cells also express Workshop Cluster 1 (WC1), which is a family of cell surface glycoproteins found in some mammals such as sheep and cattle, but absent in humans and mice [71]. These WC1+ γδ T cells are proinflammatory and produce IFN-γ upon activation [72].

Some pathogens enter the organism and grow in extracellular fluids rather than inside of cells. In this situation, a T cell-mediated response (cellular immunity) is ineffective, so it relies on antibodies produced by activated B cells (humoral immunity). Immunoglobulins (Ig) are specific molecules present on the B cell surface that recognize the antigen. These form part of the B-cell receptors (BCRs) which, after binding to these antigens in secondary lymphoid organs, promote B cell activation through T helper cells (CD4+) and differentiation into plasmatic cells, secreting soluble Ig with the same antigenic specificity. These soluble Igs become antibodies (Ab). In bovines, like in most mammals, there are five major isotypes or structural classes of antibodies or immunoglobulins based on the chains that form their molecular structure, each showing specific functions (IgM, IgG, IgA, IgD and IgE) as mentioned in Table 3. Cattle possess two IgM subclasses and three IgG subclasses: IgG1, which constitutes 50% of the total serum IgG, and is the predominant Ig in milk rather than IgA; IgG2; and IgG3. All of them have a constant region, which defines the isotype, and a variable region, which defines antigen specificity [7]. Once this region binds to an antigen, it can trigger different mechanisms to eliminate the foreign particle: opsonization, agglutination, and neutralization.

Opsonization is a process where the antigen is tagged by the antibody for its destruction. Then, phagocytic cells are drawn to where the complex is present and can engulf the foreign body. This process can be enhanced by the activation of the complement system, which leads to a cascade of proteins that lysate the antigen [74]. Agglutination occurs when multiple antigens bind to antibodies, so these start to clump together, making them more susceptible to being phagocytosed. This also reduces the number of antigens circulating within the body, avoiding dissemination and colonization [75]. Neutralization is particularly important and effective against viruses or toxin-producing pathogens. In this situation, the antibody itself prevents or suppresses the antigen or toxin activity. In fact, due to its therapeutic potential, this function has been used to design monoclonal antibodies against specific pathogen components [76].

All these events occur in a coordinated manner once the adaptive immune response is activated, leading to a proliferation of cells and antibodies prepared to efficiently neutralize pathogens and their toxins.

#### 2.2.3. Immune Memory (Trained Immunity)

Right after the complete elimination of the pathogen, effector cells undergo apoptosis, releasing TGF-β during this process to control inflammation [77]. Some memory T and B cells that proliferated during the clonal expansion will resist apoptosis and become circulating and long-lasting memory cells. These will “remember” and recognize the specific antigens that activated the immune response the first time, providing a quick and more efficient response upon following encounters [78]. The number of memory cells is normally abundant in older animals, since they are conserved throughout their life.

There are three types of memory T cells. Two of them can migrate to the site of infection: central-memory T cells (TCM), present in secondary lymphoid tissues, and effector memory T cells (TEM), present in non-lymphoid tissues. Subset differentiation depends on the expression of a chemokine receptor, CCR7. The absence of this receptor promotes TEM cells, which respond quickly upon encountering antigens and produce cytokines like IFN-γ, IL-5, and IL-4 [79]. The third type, tissue-resident memory T cells (TRM), is non-migratory and constitutes the most abundant memory T cell subset [80]. CD4+ memory T cells constituted 30% of the total CD4+ T cell population in mice, which is considerably higher than that observed for CD8+ T cells [81].

Memory B cells can be differentiated into subsets based on their immunoglobulin class, localization, and their passage through germinal centers (GCs) during clonal expansion. Germinal centers are structures within secondary lymphoid organs that allow interactions between T cells, B cells, and APCs. Here, activated B cells undergo somatic hypermutation and class-switch recombination, which will determine antibody affinity and isotype [82]. The outcome is a diverse group of activated B cells that become plasma cells or memory B cells with different properties depending on whether or not they have experienced GC reactions; an example of these properties would be producing IgM rather than IgG antibodies upon reactivation [83,84]. IgM memory B cells can initiate secondary GC responses during reinfection, while IgG memory B cells will differentiate into plasma cells capable of producing antibodies, contributing to long-term humoral immunity [85].

Other cell types can become memory cells, such as monocytes and macrophages, NK cells, and innate lymphoid cells (ILCs); these undergo epigenetic and metabolic adaptations that allow an amplified response upon reinfection [86].

## 3. Immune Responses to Pathogens

### 3.1. Viral Infections

The innate immune response usually detects viral components through endosomal TLRs, such as TLR3 (dsRNA), TLR7 (ssRNA), and TLR9 (DNA). Type I IFN (IFN-α and IFN-β) is produced after the initial encounter with viruses, promoting antiviral proteins and blocking viral replication. The following cellular response is mainly mediated by innate lymphoid cells (ILC1) lymphocytes, which produce IFN-γ and contribute to the activation of both the Th1 response and NK cells [87,88,89]. Activated NK cells also produce type II IFN (IFN-γ), which contributes to the activation of other cellular responses [7]. The adaptive immune system against viruses involves cytotoxic CD8+ T cells and B cell activation, which eliminates infected cells and produces neutralizing antibodies, respectively. The main immunoglobulins produced are IgA, which attack viruses that enter the organism through mucosal tissues; and IgG and IgM, which are both effective against virus dissemination in blood [8,90]. However, many viral pathogens have evolved strategies to evade or modulate the host immune response. Table 4 shows key findings in the immune response against these pathogens.

For example, Bovine Viral Diarrhea Virus (BVDV), one of the most significant bovine RNA viral pathogens, is known to establish persistent infections and alter immune homeostasis. Non-cytopathic (np) strains can evade innate immunity by inhibiting IFN signaling through proteins like N-terminal protease (Npro) and Envelope RNase (Erns); these downregulate antigen presentation and suppress relevant PRRs such as TLR3 and TLR7. BVDV also changes the immune profile from a Th1 to a non-protective Th2 response, reducing antigen presentation and cytokine production, which avoids a cell-mediated response and promotes secondary infections. Cytopathic strains promote apoptosis and autophagy via NF-κB modulation and DNA Damage Inducible Transcript 3 (DDIT3) induction. Overall, BVDV produces immune suppression and reduces immunity in infected cattle [91].

Similarly, Bovine Respiratory Syncytial Virus (BRSV), which is a leading cause of respiratory disease in calves, is known for modulating the immune response. Although it activates TLR3 and TLR7 pathways like other RNA viruses and induces chemokine production, it also triggers a Th2 response in the lungs. This change from a Th1 to a Th2 response is driven by altered cytokine production from pulmonary DCs, and it promotes IFN-γ suppression and impairs CD8+ activity [55,92]. BRSV also impairs neutralizing antibodies and interferes with DC-T cell interactions. This effect on the immune system contributes to weakening the immune response and memory formation, and promotes recurrent infections in adult cattle [28,93].

Other RNA viruses also display immune modulation or evasion strategies. For example, Foot-and-Mouth Disease Virus (FMDV), which causes foot-and-mouth disease (FMD). This viral disease is a highly contagious viral infection characterized by fever and vesicular lesions in the mouth, feet, and teats, leading to severe productivity losses and major economic impact on the cattle industry [118]. This virus inhibits type I IFN and NF-κB signaling via structural and non-structural proteins, which delays immune activation [105]. Chronic infections by Bovine Leukemia Virus (BLV), which causes enzootic bovine leukosis, lead to immune dysfunction via altered cytokine production, T cell exhaustion, increased IL-10 levels in plasma, and reduced APC activity, alterations which rapidly facilitate B cell transformation and lymphoma development, especially in CD5+IgM+B cell subsets [98,119].

Among DNA viruses affecting cattle, Bovine Herpesvirus type 1 (BoHV-1) is a major pathogen due to its impact on both respiratory and reproductive health. This virus is responsible for infectious bovine rhinotracheitis and has developed certain evasion strategies which target both the innate and adaptive responses. BoHV-1 interferes with IFN signaling pathways and CD8+ recognition by expressing viral proteins that effectively block MHC-I expression, disrupt immune cell transportation, and induce apoptosis in CD4+ cells [113]. Additionally, this virus promotes the degradation of STING (Stimulator of Interferon Genes, a key molecule of the innate immune system) through DDIT3 and Sequestosome 1 (SQSTM1) upregulation, which suppresses type I IFN responses at early stages of infection [109].

### 3.2. Bacterial Infections

The immune response against bacterial pathogens depends mostly on whether the bacteria are extracellular, facultative, or obligate intracellular. The immune system response against intracellular bacteria implies cell-mediated mechanisms. Initially, there is an activation of macrophages and DCs, which produce TNF-α and IL-12. Later on, the response relies on a Th1 profile, activating CD8+ T cells and secreting a range of proinflammatory cytokines such as TNF-α, IL-6, IL-1β, and IL-12 [7]. A brief summary of the normal immune response against bacterial pathogens and how these modulate or evade the immune system is shown in Table 5.

One of the most extensively studied bacteria in cattle is *Mycobacterium bovis*, a facultative intracellular pathogen responsible for bovine tuberculosis. Like other intracellular bacteria, this species involves the activation of Th1 pathways. This response includes a high production of IFN-γ, TNF-α, IL-2, IL-17, IL-21, and nitric oxide (NO) by activated macrophages [180,181]. There is a upregulation of γδ T cells, IgG2, and IFN-γ responses, while IL-4 is downregulated, which contributes to a Th1 polarization [182]. Animals infected with *M. bovis* develop memory B cells which express IgM, IgG and IgA isotypes [183], and it has been shown that PBMC cultures express Inducible Nitric Oxide Synthase (iNOS), IL-22 and IFN-γ [184]. Although this pathogen promotes strong immune activation, it employs several immune evasion mechanisms. An example of this is the inhibition of phagosome-lysosome formation and the downregulation of MHC-II expression [185] and the Kelch-like ECH-associated protein 1-Nuclear Factor, erythroid 2-like 2 (KEAP1-NFE2L2) pathway [186], and upregulation of Arginase 1 (Arg1), which ultimately reduces NO production [187]. These actions impair antigen presentation and inflammasome activity and promote the survival of bacteria inside macrophages. Additionally, this pathogen can induce a shift from a Th1 to a Th2 response and reduce CD8+ T cells, which contribute to chronic infection [188].

Another relevant bacterium in bovine management is *Brucella abortus*, which causes brucellosis, a contagious zoonoses infection characterized by reproductive disorders such as abortion, retained placenta, infertility, and decreased milk production, leading to significant economic losses in the livestock industry [189]. This intracellular pathogen involves the activation of TLR2, TLR4, and TLR9 [135], which promotes the secretion of IFN-γ, TNF-α, and IL-12 by macrophages and DCs [140]. This cytokine release enhances reactive oxygen species (ROS) and reactive nitrogen species (RNS) production, contributing to the elimination of bacteria. TLR 7 and TLR 3 can also be activated, leading to IL-6 production and type I IFN responses via the Mitogen Activated protein kinase (MPK)/NF-κB axis [136]. *B. abortus* also has some strategies to fight the host immune response. It produces low-endotoxic LPS to evade detection via TLR4 [139], and suppresses TNF-α production in early infection stages [135]. It also interferes with TLR2 activation, inhibiting maturation of DCs [143], and inhibits phagolysosome fusion via the VirB secretion system and caspase degradation, which is necessary for inflammasome activation [141].

The immune response against extracellular bacteria is initially targeted by innate mechanisms, including complement activation, which induces inflammation and contributes to the lysis of Gram-negative bacteria. Phagocytic cells recognize bacterial PAMPs through TLRs, leading to the activation and secretion of cytokines such as IL-1, TNF-α, IL-6, and IL-12, which promote leukocyte recruitment, local inflammation, and NK cell activation. Later, extracellular bacteria are controlled by humoral mechanisms, secreting antibodies (mostly IgG) that neutralize bacteria and their toxins, block adhesion to host tissues, and promote phagocytosis [4].

An example of extracellular bacteria is *Mannheimia haemolytica*, one of the most pathogenic bacteria correlated to the bovine respiratory disease complex (BRDC). This agent induces a strong innate response followed by a rapid recruitment of neutrophils and macrophages to the site of infection, triggered by its lipopolysaccharides (LPS) and leukotoxin (LktA). This potent leukotoxin selectively targets bovine leukocytes binding to CD18, which is broadly expressed in leukocyte subsets, leading them to lysis or apoptosis [156]. In addition, the LKT-LPS complex enhances this cytolytic effect and exacerbates inflammation [158]. The inflammatory response is characterized by the upregulation of proinflammatory mediators like TNF-α, IL-1β, IL-6, IL-8, prostaglandins (PG) and NO. Also, outer membrane proteins (OMPs) such as Pasteurella lipoprotein E (PlpE) and Outer membrane protein A (OmpA) contribute to immune activation and recognition by host cells [157]. *M. haemolytica* has some strategies to evade and modulate immune response. These include the inhibition of phagocytosis and lysis, suppression of neutrophil function through OMPs, and leukocyte modulation through bacterial metabolites [157]. This bacterium can also degrade key components of the adaptive immune system via hydrolysis of bovine IgG1 and specific bacterial IgA1 and IgA2 proteases that cleave bovine IgA [154,157].

Bovine-mastitis-causing bacteria are mostly extracellular, although some agents like *Staphylococcus aureus* can invade and survive within mammary epithelial cells, which means that these agents can persist in host cells. *S. aureus* is a facultative bacterium associated with subclinical and chronic mastitis in cattle. It produces a milder innate response in milk compared to other bacteria, with little increase in IL-1β, IL-12, and IFN-γ and barely any induction of TNF-α or IL-8 [120]. However, the response in mammary epithelial cells is quite different, as the pathogen is recognized via TLR2, leading to NF-κB signaling and the production of proinflammatory cytokines such as IL-1β, IL-6, IL-8, and TNF-α [121,124]. A distinctive feature of *S. aureus* is the secretion of superantigens (SAgs), which can evade conventional antigen presentation, leading to a massive, non-specific activation of 5 to 20% of the host’s T cell subsets, triggering an exacerbated Th1/Th17 profile response. Additionally, SAgs can impair the function of MAITs, γδ T cells, NK cells, B cells and mast cells [128]. *S. aureus* can also suppress host immune signaling to promote persistence in host cells. Immune evasion mechanisms include NF-κB pathway inhibition, which leads to a lower TNF-α production and downregulation of Transporter Associated with Antigen Processing (TAP), and NO- and prolactin-related signaling [126]. IκB/NF-κB (Nuclear factor Kappa-light-chain-enhancer of activated B cells) signaling is also inhibited via activation of the Wnt/β-catenin signaling pathway, which potentially affects phagocytosis and cell motility [125]. This pathogen also impairs the oxidative stress response through the inhibition of Nuclear Factor Erythroid 3-related factor 2 (Nrf2) activation via p62/SQSTM1 phosphorylation [131]. Persistent strains are also capable of delaying host inflammatory responses, with lower early cytokine production and a change in humoral response, promoting IgG2 instead of IgG1, which suggests a switch to a Th1 response [123].

### 3.3. Parasitic Infections

Parasitic infections in cattle can trigger diverse immune responses depending on the nature of the parasite, but also host factors such as age, pregnancy, productive status and system, as some of these can be transmitted through vectors or enter the organism through grazing. External parasites can impact cattle health and productivity, but significant health problems and economic losses come mostly from internal parasites such as protozoa and helminths. Bovine immune responses against some parasites are mentioned in Table 6.

Some protozoan parasites cause significant diseases in cattle. One example is *Babesia bovis*, which causes the disease bovine babesiosis, a tick-borne infection characterized by fever, anemia, hemoglobinuria, and high mortality in susceptible cattle, resulting in substantial economic losses in endemic regions [227]. This parasite is an intraerythrocytic protozoan that elicits a strong Th1 response, primarily through macrophage activation via IFN-γ and NO production [228]. This response is enhanced by RAP-1 (rhoptry-associated protein 1), a highly immunogenic protein from *B. bovis*, which also induces strong B and T cell responses [190].

It has been shown that attenuated R1A strains activate macrophages via TLR2 and promote some proinflammatory cytokines and expression of Cyclooxygenase 2 (COX-2), while virulent strains are capable of suppressing macrophage activation [193]. Interestingly, age may regulate immunity response: calves show an early upregulation of IL-12 and IFN-γ in spleen, while adults exhibit a delayed response and greater expression of IL-10 [194].

Species such as *Eimeria bovis* and *E. zuernii* are intracellular protozoa which cause coccidiosis in cattle, especially in calves. Polymorphonuclear neutrophils (PMNs) play a role in early immune defense against these pathogens by phagocytosing protozoan sporozoites. This interaction upregulates IL-8, IP-10, and IL-12 transcription and promotes the expression of proinflammatory genes such as IL-6, TNF-α, iNOS, MCP-1 and GROα [195]. It also enhances oxidative burst and NETosis, where its formation is enhanced by surface expression of the neutrophil CD11b receptor [196,229].

Another species of protozoa that causes intestinal disorders in cattle is *Cryptosporidium parvum*. This pathogen activates NF-κB via TLR2 and TLR4, producing proinflammatory cytokines like IL-6, IL-8, IL-10, IL-12 and TNF-α [211]. There is an upregulation of TLR4 in dendritic cells, as well as CD80, CD86, and MHC-II, which induce a Th-1 skewed immune response characterized by IFN-γ and IgG2 production [208,210]. *C. parvum* also triggers NET formation in neutrophils, involving key enzymes like NADPH oxidase, neutrophil elastase, and myeloperoxidase [209].

*Neospora caninum* is another protozoan parasite, and a major cause of abortion in cattle. Its infection elicits a Th1 response with IFN-γ and IL-17 production. Interestingly, this response is reduced during pregnancy, which may contribute to vertical transmissions [202]. In infected heifers, the antimicrobial peptide BMAP28 was detected in peripheral blood mononuclear cells, colostrum, and umbilical cord tissue. There was also a higher expression of TLR7 and IL-10 in the umbilical cord. Interestingly, calves present a lower IgG1/IgG2 ratio after taking the colostrum, eliciting an effective Th1 response [205]. Additionally, in vitro studies have shown that the infection activates the NLRP3 inflammasome, producing IL-1β and IL-18, and induces the cleavage of caspase-1 and cell death [207].

Helminth infections can also have a great impact on bovine health, drastically affecting their productivity. *Ostertagia ostertagi* is a significant parasitic nematode and a major cause of gastritis in cattle, which can lead to substantial economic losses. The parasite elicits an upregulation of proinflammatory cytokines like IL-4 and IL-1β [212]. Bovine neutrophils release NETs in response to the parasite, which helps to contain the infestation [213]. However, these also produce IL-10 upon exposure, which can contribute to host evasion since this cytokine downregulates the immune response [214]. Both T and B cells participate in the response, although it has been shown that products from the fourth-stage larvae (L4) can suppress T cell activation [230].

Another important helminth in bovine health is *Fasciola hepatica*, a trematode responsible for the liver fluke disease in cattle. This trematode elicits a Th2 response characterized by the production of IL-4 and IL-5, as well as an increase in eosinophils [217,218]. An altered cytokine profile is normally observed during these infections, including reduced levels of IFN-γ and TNF-α in bovine monocyte and macrophage subsets [215]. Interestingly, it has been shown that *F. hepatica* produces a greater infiltration of CD3+ T cells, CD79α+ B cells, and IgG+ plasma cells in the left hepatic lobe compared to the right lobe [216].

## 4. Conclusions

The bovine immune system represents a complex and dynamic interplay of innate and adaptive components, finely tuned to respond effectively to a wide array of pathogens. While many mechanisms mirror those observed in other mammals, particularly humans and mice, cattle exhibit distinctive features shaped by their evolutionary history, environmental exposures, and physiological demands. These species-specific adaptations underscore the need for tailored immunological models rather than relying solely on data from traditional model organisms.

The bovine innate immune response, as the first line of defense, relies on coordinated actions of epithelial barriers, pattern recognition receptors (PRRs), phagocytic cells, and inflammatory mediators. Key components, including Toll-like receptors (TLRs), NOD-like receptors (NLRs), and RIG-I-like receptors (RLRs), detect pathogen-associated molecular patterns (PAMPs) and trigger signaling cascades that lead to cytokine production, cell recruitment, and pathogen clearance. Notably, cattle exhibit unique TLR polymorphisms that influence cytokine responses and disease susceptibility, illustrating intra-species diversity and its clinical relevance. Additional bovine-specific functions in inflammasome activation and acute-phase protein expression further highlight differences in inflammatory regulation compared to other mammals.

The adaptive immune system in cattle demonstrates specialized mechanisms as well. The bovine major histocompatibility complex, BoLA (Bovine Leukocyte Antigen), shows high genetic variability affecting antigen presentation and immune responsiveness across breeds. γδ T cells, abundant in early life, contribute significantly to mucosal immunity and rapid, antigen-independent responses. Other unique features, such as MAIT cells, CD1-restricted T cells, and multiple IgG subclasses, distinguish bovine adaptive immunity and support effective responses against bacteria, viruses, and parasites, while also indicating potential limitations in immune memory and vaccine efficacy.

Infectious diseases remain a major challenge in cattle production, with pathogens such as *Mycobacterium bovis*, Bovine Viral Diarrhea Virus, *Staphylococcus aureus*, and *Fasciola hepatica* evolving strategies to evade or modulate the bovine immune response. These include the inhibition of interferon signaling, suppression of antigen presentation, modulation of T cell responses, and interference with inflammasome activity. Understanding these host-pathogen interactions is critical for improving diagnostics, vaccines, and therapeutic strategies.

Advances in understanding bovine-specific immune mechanisms are reshaping strategies for disease prevention and therapeutic intervention in livestock production. Insights into γδ T cells, WC1+ subsets, pattern recognition receptors, cytokine networks, and antibody-mediated responses are informing the development of novel vaccines, targeted immunomodulators, and precision breeding approaches to enhance disease resistance. By identifying species-specific pathways and molecular mechanisms, researchers can design interventions that are more predictive, effective, and tailored to the unique immunological landscape of cattle.

These evolving concepts not only improve herd health, productivity, and welfare but also have broader implications for veterinary and comparative immunology. Lessons learned from bovine immunity can guide the development of cross-species vaccines, enhance management of zoonotic pathogens, and inform next-generation immunotherapies in both animal and human health. Integrating genomic, proteomic, and systems-level analyses promises to accelerate the translation of immunological research into practical applications, ultimately supporting sustainable livestock production and contributing to global food security.

## Figures and Tables

**Figure 1 ijms-26-08461-f001:**
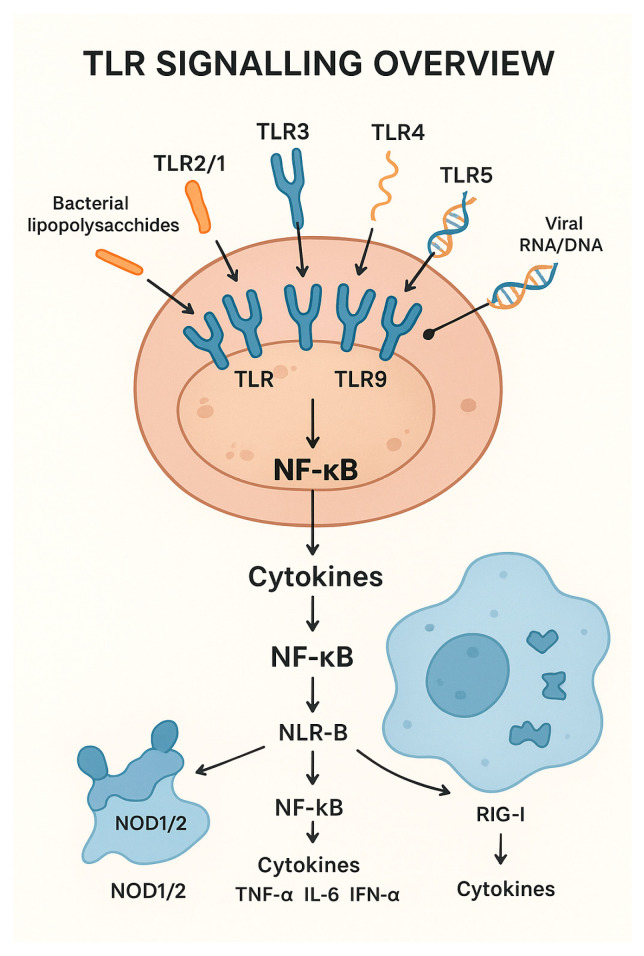
Diagram of Toll-like receptors (TLR) signaling and its relationship with cytokine activation.

**Figure 2 ijms-26-08461-f002:**
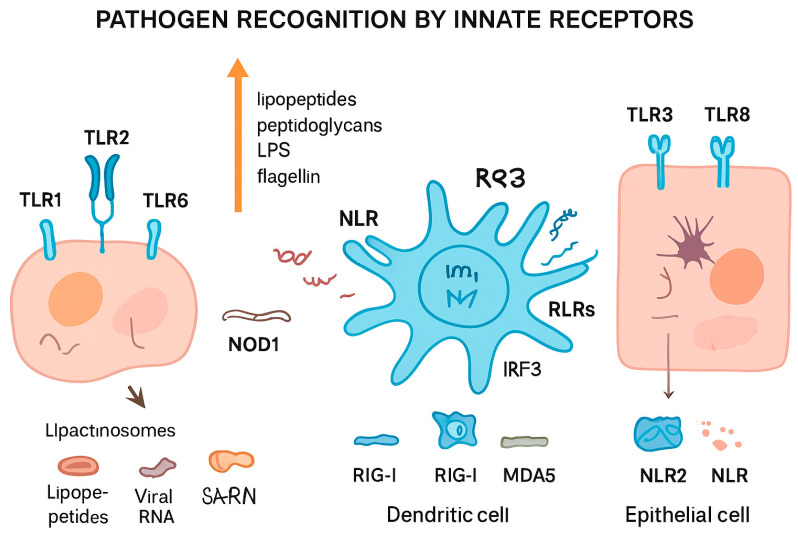
*Pathogen recognition by innate immune receptors.* Different cell types detect pathogens through specialized pattern recognition receptors (PRRs). Toll-like receptors (TLRs, e.g., TLR1, TLR2, TLR3, TLR6, TLR8) on the cell surface or in endosomes recognize microbial components such as lipopeptides, peptidoglycans, lipopolysaccharides (LPS), and flagellin, leading to the formation of lipactinosomes. NOD-like receptors (NLRs, e.g., NOD1, NLR2) detect intracellular bacterial components. Retinoic acid-inducible gene I-like receptors (RLRs, including RIG-I and MDA5) recognize viral RNA in dendritic cells, triggering IRF3-mediated signaling. This coordinated sensing allows innate immune cells, including dendritic and epithelial cells, to mount appropriate immune responses against bacterial, viral, and other pathogenic threats.

**Table 3 ijms-26-08461-t003:** Site of production, functions, and concentration in bovine serum of different immunoglobulins (Ig) [12,73].

Type	Site	Function	Serum (mg/mL)
IgM	Secondary lymphoid organs	Primary response to pathogens Complement fixation Virus neutralization Bacteria agglutination	2.5–4.0
IgG	Spleen Lymph nodes Bone marrow	Secondary response to pathogens Neonatal immunity	17–27
IgA	Body surfaces	Mucosal immunity Proteolysis protection Virus/bacteria neutralization	0.1–0.5
IgE	Body surfaces	Parasitic response	
IgD	Attached to B cells	Regulatory response	

**Table 4 ijms-26-08461-t004:** Bovine immune responses against some viral pathogens.

Type	Viral Agent	Immune Response	References
RNA	Bovine Viral Diarrhea Virus (BVDV) Cytopathic (cp) Non-cytopathic (ncp)	Increased TLR3, type I IFN, and IL-12 mRNA in monocytes 1 h post-infection (ncp), and TLR7 mRNA in monocytes 24 h post-infection (ncp, cp). IFN-β production via activation of IRF1, IRF7, and NF-κB. Proliferation of CD4+ and CD8+. Decreased MHC-I, MHC-II, and CD86 expression on Mo-DCs (ncp) and monocytes (cp). Modulation of PRRs; inhibition of type I IFN via Npro, Erns, NS4B, and DDIT3; suppression of APC function and MHC expression; alteration of NF-κB; autophagy/apoptosis induction (Cytopathic BVDV); evasion of innate and adaptive immunity.	[91]
	Bovine Respiratory Syncytial Virus (BRSV)	γδ T cells response through TLR3 and TLR7 ligands; production of chemokines CCL2 and CCL3. Inhibition of CD8+ T cells and IFN-γ responses by promoting Th2 polarization and IgE production. Long-lasting humoral immunity but weak mucosal and no detectable T cell responses. Immunity evasion via NS1/NS2 inhibition of type I IFN, NF-κB sequestration, Th2/IL-17 polarization, DC dysfunction, and secretion of decoy G protein (sG), which suppresses effective CD8+ and Th1 responses.	[28,55,92,93]
	Bovine Parainfluenza Virus Type 3 (BPIV-3)	CD4+ and CD8+ cells and Th1 cytokine production. TNFα, IL1β and IL6 upregulation upon infection. Infected alveolar macrophages suppress lymphocyte proliferation via cell contact and abortive infection, impairing adaptive immune response. Activation of p38 MAPK via MKK3 upregulation in host cells, enhancing viral replication and contributing to immunosuppression.	[94,95,96,97]
	Bovine Leukemia Virus (BLV)	CD5+ IgM+ B cells transformation via Tax; evasion of CTL responses by limiting Tax expression. Cytokine imbalance, T cell exhaustion, altered B cell and monocyte function, reduced antibody and phagocytic responses, impaired MHC-II response. Suppression of IgM production via downregulation of BLIMP1 and BCL6 in IgM+ B cells. Alteration of type I IFN signaling and antiviral gene expression in mammary epithelial cells. Treg/Breg-driven immune suppression in cows with persistent lymphocytosis. Cytokine expression alteration in blood and milk; high proviral load in PBMCs (peripheral blood mononuclear cells) associated with lower IL-12/IL-6 and increased IL-10, contributing to mastitis susceptibility. Immune suppression and upregulation of proliferation genes.	[98,99,100,101,102,103,104]
	Foot-and-Mouth Disease Virus (FMDV)	Immune evasion via structural proteins (VP0–VP4) targeting IRF3, VISA, JAK–STAT and autophagy. Immune evasion via non-structural proteins (Lpro, 2B/3A/3C) that degrade PRRs and block IFN, NF-κB, autophagy and JAK–STAT signaling.	[105,106]
	Bluetongue Virus (BTV)	Induction of type I IFN and TNF-α, IL-1β, IL-8, CCL2 and E-selectin. Induction of Th2 immune response (IL-4, IgE) in PBMCs. Suppression of innate and Th1 cytokines (TNF-α, IFN-γ, IL-12). Inhibition of type I IFN signaling via NS3/NS4/VP3.	[107,108,109]
DNA	Lumpy Skin Disease Virus (LSDV)	IFN-γ production and neutralizing antibodies (IgM followed by IgG). Activation of CD4+ and CD8+. Increased IFN-γ, TNF-α, decreased IL-10. ORF127 inhibits IFN-β via TBK1 (less ubiquitination and phosphorylation); blocks cGAS-STING.	[110,111,112]
	Bovine herpesvirus type 1 (BoHV-1)	Antibodies (anti-gB/gC/gD/gH), CD8+ activation, inflammasome formation. Inhibition of IFN responses, CD8+ and CD4+ T cell function and cell-mediated immunity. bICP0 inhibits IFN-β via IRF3 degradation and disrupts PML bodies. bICP27 reduces IFN-β1/β3 expression via TBK1-STING interference. gG binds chemokines, impairs immune cell trafficking. gN blocks TAP and impairs MHC-I presentation. VP8 inhibits STAT1 nuclear translocation and blocks IFN-γ/α/β signaling. Inhibition of IFN-β, promoting MAVS ubiquitination and impairing relation between IRF3 and CBP/p300. Tegument protein UL3 inhibits type I IFN by promoting STING degradation; enhanced ATG101 expression; strengthens ATG101–STING interaction.	[113,114,115]
	Bovine Papillomavirus (BPV)	RIG-I and MDA5 activation; MAVS-TBK1-IRF3 signaling and type I IFN production. MHC-I, TRIM25, RIG-I, MDA5 and Sec13 downregulation through E5, reducing IRF3 activation and type I IFN response.	[116,117]

**Table 5 ijms-26-08461-t005:** Bovine immune response against bacterial pathogens. NO: nitric oxide; IL: interleukin; TFN: Tumor Necrosis Factor; IFN: Interferon; TLR: Toll-like receptor; IRF: interferon regulatory factor; NOS: Nitric oxide synthase; PBMC: Peripheral Blood Mononuclear Cell.

Type	Bacterial Agent	Immune Response	References
Gram +	*Mycobacterium bovis*	Th1 response, macrophage activation, NO production, γδ T response, IgG2 production. Upregulation of IL-1α, IL-1β, IL-10, IL-17α, TNF-α, IFN-γ, IL1R1, TLR2, TLR4, IRF5 and Arg1. Downregulation of MHC-II and IL-4. Inhibition of phagosome-lysosome fusion. Memory B cells of IgM, IgG, and IgA isotypes. iNOS, IL-22, and IFN-γ expression in bovine PBMC cultures. Mb04-303 (avirulent strain) induces type I IFN signaling, greater phagosome membrane damage. Shift from Th1 to Th2 response, CD8+ reduction, atypical memory B cells. KEAP1-NFE2L2 pathway downregulation, impairing antigen presentation, inflammasome activity and reducing IL-1β production.	[119,120,121,122,123,124,125,126,127,128]
	*Staphylococcus aureus*	TLR-2 activation, IL-1β, IL-6, IL-8 and TNF-α production. Mild IL-1β, IL-12 and IFN-γ production and no TNF-α or IL-8 in milk. NF-κB pathway involvement in mammary epithelial cells. SAgs activate 5-20% of T cells by binding to MHC II and TCR Vβ regions, triggering massive Th1/Th17 cytokine production; APCs stimulation; MAIT, γδ T, NK, B and MAST cells activation. Possible modulation of APCs via TLR2/NOD1 pathways. GTB/ST8 strain secretomes enhance PBMC promotes IL-1β, STAT1 and miR-155-5p expression. NF-κB inhibition, reducing TNF-α, TAP, NO and PRL signaling. Decreased α5β1 integrin levels, reduced immune response via PRL, AP-1 inhibition. IκB/NF-κB signaling inhibition via wnt/β-catenin activation; promotion of cytoskeletal rearrangement through Rho GTPases. Impairs Nrf2 activation via p62/SQSTM1 phosphorylation. A persistent strain showed delayed inflammation, reduced early cytokine release (IL-1β, IL-6) and promoted IgG2 response. SAgs promote IL-10, PD-L1, IDO and induce apoptosis in monocytes/macrophages via TNF-α; impaired γδ/MAIT responses, neutrophil and complement function without T cell activation. GTS/ST398 strain secretome drastically reduces PBMC viability. Disruption of late endosomes and lysosomes through an effector of Rab11.	[120,121,122,123,124,125,126,127,128,129,130]
	*Streptococcus agalactiae*	Upregulation of granulocyte adhesion (ST103) and Th1/Th2 cell activation (ST12). ROS and NETs production at low multiplicity of infection. IL-1β, IL-8, IL-12β, and TNF-α production in milk, local immune response in udder. Downregulation of phagosome formation (ST103) and complement activation (ST12). Suppression of ROS, NETs, and neutrophil necrosis at high multiplicity of infection; immune evasion via cytotoxicity. IL-10 and TGF-β activation.	[131,132,133,134]
Gram −	*Brucella abortus*	MIP-1α/β, IL-8, and RANTES upregulated. TLR2, TLR4 and TLR9 activation. IFN-γ, IL-12, and TNF- α secretion, macrophage activity via ROS/RNS, and naive CD4 T cell stimulation; DCs maturation dependent on caspase-2 and TLR6. DCs activation via TLR7 (IL-12) and TLR3/TLR7 (IL-6, IP-10), promoting MPK/NF-κB and type I IFN activation. Production of low-endotoxic LPS to escape TLR4. TNF-α suppression early in infection reduced PAMP expression. Decreased TLR-dependent response. DCs maturation inhibition via TLR2 interference. UPR via STING and c-di-GMP, promoting IFN-β. Phagolysosome fusion evasion via VirB; production of low-endotoxic LPS to escape TLR4; TcpB production to inhibit NF-κB; caspase degradation; reduced IL-1β; MHC-I and MHC-II inhibition.	[135,136,137,138,139,140,141,142,143]
	*Histophilus somni*	Neutrophil and macrophage extracellular traps (ETs) formation. IgG2 production; limited cell-mediated immunity. Resistance to NO levels. Inhibits superoxide anion (O2−) produced by alveolar macrophages and neutrophils. May escape ETs through degradation via DNases. LOS, Ig-binding proteins, MOMP, and other OMPs interfere with immune detection and phagocyte inhibition; induce macrophage death; and cause antigenic variation.	[144,145,146,147]
	*Salmonella* spp.	*S. Typhimurium* in the bovine intestinal mucosa activates various signaling pathways, including MAPK, mTOR, and TGF-β. Increased CD14 and CD18, phagocytosis, and oxidative burst in neutrophils and monocytes upon exposure to different serovars. Cytokine production differs among serovars: highest TNF-α in *S. Enteritidis* and *S. Typhimurium*; IL-8 in *S. Dublin*. DCs and macrophages are key players in the immune response, although macrophages exhibit a stronger inflammatory response (IL-1β, IL-6) compared to DCs. *S. Dublin*-infected cells show elevated levels of MHCII and CD40.	[148,149,150,151]
	*Campylobacter* spp.	Neutrophil surface activity and phagocytosis enhanced by IgG. Transient proinflammatory response (IL-1β, IL-8); increased IFN-γ over time. Lack of IgA-mediated opsonization and TNF-α response.	[152,153]
	*Mannheimia haemolytica*	Macrophage and neutrophil activation via LPS and leukotoxin (LKT); increased TNF-α, IL-1β, IL-6, IL-8, PG, and NO; membrane proteins like PlpE, and OmpA contribute to immune activation. LKT binds to CD18 receptor and has a cytolytic effect on leukocytes. Phagocytosis and lysis inhibition; LPS-LKT complexes enhance cytotoxicity; neutrophil suppression via OMPs; leukocyte modulation via metabolites; IgG1 hydrolysis. Bacterial IgA1 and IgA2 proteases cleave bovine IgA.	[154,155,156,157]
	*Pasteurella multocida*	LPS detection via TLR4; NF-κB activation; IL-1α, IL-6, TNF-α, IFN-γ, and IL-12 production. TLR4 and NF-κB activation, TNF-α, IL-1β, IL-8, and NO production, neutrophil/macrophage recruitment, IgA and IgG response. NETs formation; iNOS induced in lung cells. IgA and IgG proteases, capsule interference with phagocytosis and complement, LPS/LKT mediated leukocyte lysis. LPS can trigger leukocyte lysis via mitochondrial dysfunction and caspase activation.	[158,159,160,161]
	*Escherichia coli*	TNF-α, IL-1β, IL-8, IL-12, IFN-γ and IL-10 production; early neutrophil and complement activation (C5a); augmented sCD14 and LBP; TLR4-mediated signaling. Strong TLR2/4 upregulation, NF-κB activation, and cytokine release (TNF-α, IL-1β, IL-6, IL-8). Strain-dependent differences in cytokine production (TNF-α, IL-6, IL-17). Gb3/CD77 receptor expression in lymphocytes and epithelial cells. Neutrophil recruitment; APPs secretion. TNF-α, IL-6, MCP-1, MIP-1α/β, IP-10, LBP, Hp, and SAA production. Unlike milk, colostrum LPS-IgG complexes inhibit LPS endotoxic activity and ROS production. Recognition via TLR2, TLR4, and NLRP3; MAPK and NF-κB activation; PGD2 and IL production. IL-10 limits TNF-α and IFN-γ; sCD14+ LBP enhances detection but also regulates response. Chronic infection implies reduced IRF1, CD83, IL-1α, IL-6, IL-8 and CCL20 expression than acute infection. TLR4 downregulation; strain-specific immune modulation. Shiga toxins (Stx) bind to Gb3+ cells, blunt T cell activation.	[120,162,163,164,165,166,167,168,169]
	*Leptospira* spp.	NETs formation; ROS, RNS, IL-1β, IL-8, MIP-1α and TNF production. LPS stimulates IL-1β, TNF-α and NO-mediated apoptosis. High IL-6 levels in infected uterine tissues. Does not trigger TLR-mediated inflammation in endometrial cells. TLR2 activation (not TLR4); inflammasome activation without causing pyroptosis.	[170,171,172,173]
Lacks cell wall	*Mycoplasma bovis*	Moderate IL-1β, IL-6, IL-8, TNF-α, TLR2 and TLR4 upregulation in bMECs. Humoral and cell-mediated immunity; increased CD4+, CD8+ and γδ T-cells and B cell proliferation; IgG1 and IgA production; mixed Th1/Th2 profile; high IL-2, IL-4, IL-12 and low IFN-γ. Increased IL-36A, IL-27, IFN-γ, IL-17, BATF and SLAMF1/7 transcription in PBMCs. Increased IL-1β, IL-6, IL-8, IL-12p40, and IL-17A expression in synovial tissues and fluid. Macrophage METs via NADPH oxidase-dependent ROS production. Weaken Th1 response; altered cytokine production; opsonization evasion; biofilm formation; Ig degrading proteins (MIB/MIP); phagocyte impairment and TLR interference; TNF-α and IFN-γ suppression; suboptimal IL-1 response. MET formation leads to cell lysis independent of apoptosis.	[174,175,176,177,178,179]

**Table 6 ijms-26-08461-t006:** Bovine immune system against some parasitic pathogens.

Type	Agent	Immune Response	References
Protozoa	*Babesia bovis*	Macrophage activation via IFN-γ and parasite derived products, NO production. IgG2 antibodies production via IFN-γ. RAP-1 (rhoptry-associated protein 1) induces strong B and T cell responses, produces IFN-γ and a dominant Th1 response. Early activation of IL-12 and IFN-γ transcripts in the spleen in calves; this response is delayed in adults, which showed IL-10 expression. High CD8+ T cell expression in the spleens of both calves and adults. Attenuated R1A strains activate macrophages via TLR2, COX-2 expression, and proinflammatory cytokines. VP2 strains may evade innate immunity by suppressing macrophage activation.	[190,191,192,193,194]
	*Eimeria* spp.	Polymorphonuclear neutrophils (PMN) interact with sporozoites via engulfment or filopodia and upregulate proinflammatory genes (IL-6, TNF-α, iNOS, MCP-1, GROα). PMN triggers NET formation (enhanced by CD11b expression), oxidative burst, and phagocytosis. Increased transcription of IL-8, IP-10, and IL-12. CD4+ and CD8+ are involved. γδ T cells also participate, especially during primary infection. Upregulation of IFN-γ and IL-2 (CD4+) during primary infection.	[195,196,197,198]
	*Tritrichomonas foetus*	IgG2 antibodies present in serum, IgG1 antibodies in vaginal mucus. Neutrophils rapidly kill the parasite via trogocytosis. Apoptosis induction in bovine vaginal epithelial cells via cysteine protease 30 (CP30) release.	[199,200,201]
	*Neospora caninum*	Th1 response with IFN-γ and IL-17 production. This response is reduced during pregnancy. NLRP3 inflammasome activation with production of IL-1β and IL-18. CD4+, CD8+, γδ T, and NK cells involvement. IgG antibodies observed in infected heifers and their colostrum. BMAP28 is in peripheral blood mononuclear cells, colostrum, and umbilical cord. TLR7 and IL-10 expression in the umbilical cord. IgG2 in calves after taking the colostrum.	[202,203,204,205,206,207]
	*Cryptosporidium parvum*	Upregulation of TLR2 and TLR4 upon infection. NF-κB activation and production of IL-6, IL-8, IL-10, IL-12, and TNF-α. Enhanced expression of TLR4, CD80, CD86, and MHC-II in DCs, promoting a Th1 response with IFN-γ and IgG2 production. NETs formation in neutrophils, involving NADPH oxidase, neutrophil elastase, and myeloperoxidase.	[208,209,210,211]
Helminths	*Ostertagia ostertagi*	IL-4 and IL-1β upregulation. Neutrophils release NETs and produce IL-10. Products from the L4 can suppress T cell activation.	[212,213,214]
	*Fasciola hepatica*	Th2 response with IL-4 and IL-5 production, and an increase in eosinophils. Altered cytokine profiles, including reduced IFN-γ and TNF-α in monocytes and macrophages. Greater infiltration of CD3+ T cells, CD79α+ B cells, and IgG+ plasma cells in the left hepatic lobe.	[215,216,217,218]
	*Dictyocaulus viviparus*	Specific antibody response, including IgG1 and IgE. Upregulation of IL-2, IL-4, IL-5, IL-10, IL-12p35, IL-13 and IFN-γ.	[219,220,221]
Arthropods	*Hypoderma lineatum*	CD4+ T cells are predominant during primary infestation; CD8+ and CD3+ T cells increase at 48 h post-infection (hpi). Both Th1 and Th2 cytokines are involved. IL-4 levels increase in reinfested individuals. IFN-γ increases at 6 hpi, while IL-10 peaks at 48 hpi. Hypodermin B (HB) reduces IFN-γ expression and stimulates IL-10. IgG1 production (especially in presence of warbles); IgG2 and IgM levels show irregular patterns.	[222,223,224,225,226]

## Data Availability

Not applicable.

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
