# Peer review of "A Comprehensive Review of the Bovine Immune Response to Pathogens"

_ijms, 2025, doi:10.3390/ijms26178461_

Round 1
Reviewer 1 Report
Comments and Suggestions for Authors
The review offers a clear and updated body of knowledge on both innate and adaptive immune responses in cattle and is well designed. Each section is structured logically and contributes effectively to the overall understanding of the topic.
Every research study related to the main topic is well explained and supported by precise and timely references. The review demonstrates a comprehensive grasp of the current literature and integrates it meaningfully.
Each paragraph is well-focused and detailed, with appropriate use of citations.
A few minor suggestions are noted:
- add some citations in the last section of introduction (from line 38 to line 59 there aren't citations)
Tables are clearly presented and logically structured. However, a few improvements are recommended:
- add some figures to improve the clarity of the text
It would be beneficial to expand the Conclusion section by discussing the implications of evolving concepts in therapeutic strategies and disease prevention in livestock production.. Emphasizing the potential impact of such research on the broader field would strengthen the closing message.
Author Response
Read the attach file
Reviewer 2 Report
Comments and Suggestions for Authors
The study by Lesta A., and colleagues presents a review of the immune system of bovines. The main weaknesses of this review are the lack of specific information and the way the information is presented in the introduction, particularly in the section on the bovine immune system (Part 2). This manuscript would benefit if, within each section on innate and adaptive immunity, the authors first indicate which proteins and receptors are present in the bovine immune system. While it is understandable that the authors begin by explaining the overall functioning of the immune system, they should then specify the components particular to bovines. Unfortunately, after reading Section 2 (Bovine immune system), it is still unclear which cytokines, NLRs, RLRs, inflammasomes, APPs, plasma proteins, and other molecules are present in bovines. From the text, it appears that Tables 1 to 3 describe the general functions of these proteins, rather than being specific to bovines. In addition, in Section 3 the authors sometimes refer to cell types that are not mentioned in Section 2, which makes it difficult to follow. Finally, to improve the introduction, it would be preferable for the authors to first indicate which proteins are present and already identified in this group, and then explain their functions.
In section 3 for some of the pathogens the authors indicate that they are responsible for in bovines, however, this is not consistent for all the pathogens they mention, and is important.
- Other text correction:
- Several sentences/paragraphs need references;
- Line 34 and 35; 47 to 53; 163; 208; 209-211; 246-257; 271-272; 273-276; 302-304; 312-315; 357-359;412-417
- Authors should first define the term in full, followed by its abbreviation (please revise the entire text)
- Lines 295, 297, 339, 461, 484, 487, 488, 521, 542, 543, 545, 563,
- As they did in other tables, the authors should insert references in Tables 1 to Table 3.
- The names of pathogens should always be in italic (please revise the entire text)
- The authors should uniformize the use of “a”/“α” and “b”/”β”, like in IL-1b/IL-1β. There are other situations
- The authors indicate some abbreviations, but do not indicate what they mean:
- Line 340 – WC1
- Line 431 – Npro and Erns
- Several sentences/paragraphs need references;
- Line 435- DDIT3
- Line 463 – SQSTM1
- Line 484 – iNOS
- Line 487 – KEAP/NFE2L2
- Line 488 – Arg1
- Line 498 – MPK
- Line 521 – PlpE and OmpA
- Line 542 – TAP
- Line 543 – IkB/NF-kB amd Wnt/n-catetin
- Line 545 -Nrf2
- Line 565 – R1A
- Line 566 – COX
- There are many others, please revise the entire text.
Author Response
read the attach file

Round 2
Reviewer 2 Report
Comments and Suggestions for Authors
The study by Lesta A., and colleagues presents a review of the immune system of bovines. Here are some suggestions to the review version:
Introduction:
- Thanks for adding that information. However, I suggest dividing the paragraph inserted in Section 2.1 and 2.2 and relocating the relevant parts into the sections where they are specifically mentioned, rather than keeping them all together. Below, I provide some suggestions—please apply the same approach to all other components in section 2.1 and 2.2, respectively. The main goal is to indicate which of these components are present in bovines.”
- Line 69 - 93 – Instead of “Innate immunity serves as the first line of defense in bovines, providing rapid recognition and response to pathogens. Central to this system are Pattern Recognition Receptors (PRRs), including Toll-like receptors (TLRs), NOD-like receptors (NLRs), and RIG-I-like receptors (RLRs). TLRs (TLR1–TLR9) are expressed in macrophages, dendritic cells,and epithelial cells, detecting bacterial, viral, and parasitic components, while NLRs (e.g., NOD1, NOD2, NLRC4, NLRP3) mediate intracellular pathogen sensing and inflammasome formation. RLRs, including RIG-I and MDA5, recognize viral RNA within the cytoplasm, initiating antiviral responses. Inflammasomes such as NLRP3 and AIM2 play key roles in inflammation by activating caspase-1, leading to the secretion of IL-1β and IL-18, and detecting cytosolic DNA, respectively. Complementing these pathways, acutephase proteins (APPs) including haptoglobin, serum amyloid A, fibrinogen, C-reactive protein, and alpha-1-acid glycoprotein are predominantly produced by the liver during infection, contributing to pathogen opsonization, complement activation, and modulation of inflammatory responses. Cytokines orchestrate innate immune signaling, with pro-inflammatory mediators (TNF-α, IL-1β, IL-6, IL-12, IFN-γ) promoting pathogen clearance, and anti-inflammatory cytokines (IL-10, TGF-β) maintaining immune homeostasis. Phagocytic cells—macrophages, neutrophils, and dendritic cells—play complementary roles, from direct pathogen engulfment and reactive species production to antigen presentation and activation of adaptive immunity. Innate lymphoid cells (ILCs) and natural killer (NK) cells further enhance host defense. Classical NK cells mediate cytotoxicity and secrete IFN-γ, while bovine NKT cells, including WC1+ γδ T cells, bridge innate and adaptive immunity. Together, these cellular and molecular components establish a dynamic and efficient innate immune network, enabling bovines to detect, respond to, and control diverse pathogens. The innate immune response involves three relevant stages.” May be authors consider to replace by “Innate immunity serves as the first line of defense, providing rapid recognition and response to pathogens. The innate immune response involves three relevant stages…”
- Line 126 – Could be “Pattern Recognition Receptors (PRRs), including Toll-like receptors (TLRs), NOD-like receptors (NLRs), and RIG-I-like receptors (RLRs) are central components of the innate immune system. TLRs play an important role in pathogen recognition. These proteins, located on the surface of different immune cells, can detect PAMPs. When TLRs are activated, they start a chain reaction inside the cell that ends with the production of inflammatory molecules, such as cytokines [13,14]. There have been ten functional TLRs identified in bovines (TLR1 to TLR10) that are expressed in macrophages, dendritic cells, and epithelial cells, detecting bacterial, viral, and parasitic components (add REFERENCE). For example, TLR2 forms …”
- Line 145 to 152 – Could be “NLRs are proteins present in macrophages, dendritic cells, and epithelial cells. These are found exclusively in the cytosol and the nucleus, facilitating the detection of foreign nucleic acids and other intracellular components inside the cells. In bovines there been identified x “please indicate the number of NLRs identified so far” NLRs (please indicate the NLRs identified). NOD1 and NOD2 can activate NF-κB pathway…”
- Line 152 - RLRs can detect foreign RNA within the cytoplasm, initiating signaling and activating transcription factors such as NF-κB and IRF-3 (interferon regulatory factor 3) [27]. In Bovines two RLRs have been identified (RIG-I and MDA5 –“please confirm if is this”)
- Please confirm and indicate in the legend if the Table 1 indicates the main cytokines in mammals, vertebrates or if they are specific to bovines.
- Please confirm and indicate in the legend if the Table 2 indicates the main Th cell subtypes in mammals, vertebrates or if they are specific to bovines.
- Please confirm and indicate in the legend if the Table 3 indicates the Ig in mammals, vertebrates or if they are specific to bovines.
- The figures added are really nice and informative.
- My previous comment, “In section 3 for some of the pathogens the authors indicate that they are responsible for in bovines, however, this is not consistent for all the pathogens they mention, and is important.” Was not clear to the authors. What I meant is that all pathogens mentioned in the text should include information about the diseases they cause in bovines. For example, in lines 539-540 “One of the most extensively studied bacteria in cattle is Mycobacterium bovis, a facultative intracellular pathogen responsible for bovine tuberculosis….” the pathology is indicated. However, the same information is missing for Brucella abortus, Babesia bovis, and Foot-and-Mouth Disease Virus (FMDV).”
Author Response
read the attached pdf
